# An Experimental Comparison of Simple Measurements Used for the Characterization of Sand Equestrian Surfaces

**DOI:** 10.3390/ani11102896

**Published:** 2021-10-05

**Authors:** María Alejandra Blanco, Raúl Hourquebie, Kaleb Dempsey, Peter Schmitt, Michael (Mick) Peterson

**Affiliations:** 1Facultad de Ingeniería y Ciencias Agropecuarias, Pontificia Universidad Católica Argentina, Buenos Aires 1300, Argentina; 2Escuela de Ingeniería y Ciencias Agroalimentarias, Universidad de Morón, Morón 1708, Argentina; rhourquebie@gmail.com; 3Racing Surfaces Testing Laboratory, Lexington, KY 40502, USA; kalebdempsey.rstl@gmail.com (K.D.); Peter.Schmitt@uky.edu (P.S.); 4Biosystems and Agricultural Engineering, University of Kentucky, Lexington, KY 40503, USA

**Keywords:** equine, arenas, sand, base layers, portable tools, safety, equine welfare

## Abstract

**Simple Summary:**

Consistency of equestrian surfaces can contribute to safety and performance. An optimal surface is influenced by the design and material selection as well as maintenance and climate. To improve surfaces the quantitative testing of functional surface properties must expand beyond the current testing at the highest levels of competition. More widespread quantitative measurements would have a positive influence on animal welfare and rider safety. To expand beyond the current top levels of the sport, simple tools are required that can be shown to detect relevant changes in construction and maintenance. Our work suggests that the appropriate use of simple devices can help with both quality control of new surfaces and the monitoring of existing surfaces. Performance modifications to the layered surface design and addition of Geotextile were detected using the Going Stick and a simple impact test. These measured results are also influenced by other factors related to the surface condition such as moisture. Caution must be exercised in the interpretation of the results since these tools have not been demonstrated to correlate to either performance or safety of the surface. However, these results are encouraging and provide a justification for future development of this type of equipment.

**Abstract:**

Quantitative measurements of performance parameters have the potential to increase consistency and enhance performance of the surfaces as well as to contribute to the safety of horses and riders. This study investigates how factors known to influence the performance of the surface, incorporation of a drainage package, control of the moisture control, and introduction of a geotextile reinforcement, affect quantitative measurements of arena materials. The measurements are made by using affordable lightweight testing tools which are readily available or easily constructed. Sixteen boxes with arena materials at a consistent depth were tested with the Going Stick (GS), both penetration resistance and shear, the impact test device (ITD), and the rotational peak shear device (RPS). Volumetric moisture content (VMC %) was also tested with time–domain reflectometry (TDR). Results obtained using GS, RPS, ITD, and TDR indicate that the presence of the drainage package, moisture content, and geotextile addition were detected. Alterations due to combinations of treatments could also be detected by GS, ITD, and TDR. While the testing showed some limitations of these devices, the potential exists to utilize them for quality control of new installations as well as for the monitoring of maintenance of the surfaces.

## 1. Introduction

Recent research has considered the effect of equestrian surfaces on the incidence of injuries [1,2,3,4]. In addition, the surface also has an impact on the performance of the horse [5,6]. As a result, quantitative measures of surfaces have now been embraced by the International Equestrian Federation (FEI) [7] as well as North America horse racing. The use of quantitative measures is particularly promising in horse racing where the potential exists to link these measures to the extensive epidemiological data available [8,9,10]. The quantitative measures developed by the FEI initiatives include five different functional properties, firmness, cushioning, rebound, grip, uniformity, and consistency [11]. In order to maintain optimal levels of these functional properties, consistent depth of the surface, properly classified sand, consistent moisture control, proper maintenance of the surface and the appropriate selection of materials used for the base and sand additives are generally accepted as having a direct impact on the horse [12,13,14]. These inputs change the vertical and horizontal deceleration of the hoof and the resulting forces on the limbs of the horse.

The construction of surfaces for both training and competition should be based on an understanding of equine biomechanics. Therefore, to fully characterize an equestrian surface, it is necessary to replicate both horizonal and vertical loads and using a loading rate that matches the equine athlete. Both the rate and load are important since arena construction materials include unsaturated particulate materials which are in general both non-linear and strain rate dependent [15]. However, the load and load rate also depend on the riding discipline, the specific action being performed and the gait. Therefore, it is necessary to identify those portions of each discipline and those portions of each routine which are critical to the performance and safety of the athletes [12,16,17,18].

Measurement of these parameters with instruments which mimic the loads and speed of the horse are ideal [12,19,20]. However, the costs associated with the size and difficulties in transporting the instrument presents a significant barrier for acceptance outside of the most elite portion of the sport. It is also important that the data resulting from measurements are easily interpreted by practitioners rather than researchers. However, directly reducing the size or speed of the loading is not possible since the speed and size of the equine athlete is fixed. The objective of this work is to determine whether some of the aspects of construction which are known to affect performance can be detected with smaller tools. If these smaller instruments can be used for quality control, then the potential exists to expand the use of quantitative measurements to surfaces and regions where cost is a bigger constraint. Ideally, these smaller tools could be locally fabricated to further reduce cost. A low cost tool that can be easily transported for evaluation of surfaces which makes use of commercially available data acquisition has the potential to key barriers to adoption. Simple tools constructed by the arena builders and owners can also expand awareness of proper maintenance and design. 

### 1.1. Motivation for Horizonal and Vertical Measures

Two of the functional parameters described by the FEI research, cushioning and firmness, are concerned with the vertical response of the surface. These functional parameters relate to the initial surface impact and the secondary loading from the mass of the body of the animal dynamically transferring to support from the leg. Shear, the horizontal component of the load on the surface, is in one direction during the first and second impact and then reverses during breakover of the hoof as the surface supports propulsion [12]. Shear resistance affects the extent to which the hoof will slide back and forth or rotate on landing, turning, pushing off, passage, pirouette, or in a sudden brake [21,22]. Longitudinal traction affects the sliding of the hoof in the horizontal plane when braking in a linear movement. It also affects the resistance of the surface to penetration by the hoof in the form of the angle of the hoof to the ground, during breakover or the penetration of the surface by the inside portion of the hoof in a sharp turn. The limb of the horse rotates about the horizontal axis of the hoof or limb which is resisted by friction between the particles and reinforcing fibers in the equestrian surface [12,21]. The current state of the art for the testing of surfaces is the Orono Biomechanical Surface Tester which is used in both show jumping [5] and in racing surfaces [19,20]. The simplified tools chosen to characterize the surface must include not only vertical loading characteristics but also the shear behavior since these characteristics may not be correlated between surfaces. 

### 1.2. Proposed Smaller Tools

To characterize both the vertical and horizontal response of the surface, either separate tools are required or tools with more than one axis of measurement are required. For turf surfaces, a strain gauge-based sensor system, the Going Stick (Turftrax Ltd., Cambridgeshire, UK), is widely accepted in Thoroughbred racing [23] and has been proposed as an international standard [24]. This tool measures both penetration resistance and the resistance to shearing of the surface. These two measures are acquired from pushing the device into the surface and then rotating it 45 degrees. It is reasonable to assume that these two motions are related to the impact and propulsion response of the surface, even if the measurements are distinctly different in their action. The peak value of deceleration for a small mass dropped from a low distance is used in some sports applications and is commonly based on ASTM D5874-1 [25]. A rotational shear tester described by ASTM F2333-04 [26] is a surface measurement of rotation unlike the Going Stick which measures the resistance to shear below the surface. Both of the devices associated with the ASTM methods use simple low-cost hardware. It is also possible to make use of simple low-cost data acquisition for both of these devices. Since surface moisture content controls the response of nearly all of these tools, a moisture probe based on an older ASTM standard D6565 is also included in the work. In addition to the low cost of the tools described in the three ASTM standards, all of them are also easily transportable. 

With the exception of the Going Stick, the loads used in these measurements is low. This is useful for evaluating the condition of the top layer of the surface and may be important for quality control of the surface installation. In addition, the compaction which results from repeatedly dropping a small mass on the surface will be influenced by additives in the surface and the selection of sand. In fact, the original intent of the device used in the vertical impact is to evaluate the level of compaction and stability of the base materials used for a road or building foundation [27,28]. While these lightweight devices are generally better suited to describing static loading of the surface and loading by animals much smaller than a horse, they may make it possible to infer the condition of the deeper layers of the track. They are useful for the evaluation of each of the layers when surfaces are installed in multiple layers. 

### 1.3. Other Small Devices

Other lightweight devices that measure the resistance to penetration by a conical probe have also been investigated. These devices, normally referred to as penetrometers, can either be devices with a smaller probe which measure dynamic response [29] or large truck-mounted devices [30] or the smaller handheld quasi-static devices commonly used in agriculture [31]. All these penetrometers would potentially test the vertical maximal peak load for the deeper layers of the surface independent of the horizontal surface response. None of these devices replicate the dynamic loading where the vertical and horizontal peak load are related [12]. An experimental design should also include drops at a series of depths to get information related to the response of the profile [13,32]. The other devices also have a range feature that are inconvenient features for daily measurements, such as pivoting the rod at each drop, variation in the applied vertical forces depending on the depth and the effect on the surface compaction of the wheels supporting the penetrometers. Like the Going Stick, these devices are sensitive to compaction of the lower layers of the surface which influence the loading of the limb at higher speeds, such as during a gallop or a jump landing. The large dynamic loads under these conditions, which are as much as 2.5 times body weight, are significantly influenced by the properties of the deeper layers of the surface [12]. However, since most of these devices are similar to the penetration measure from the Going Stick, they would not be expected to add a unique parameter to the study. The Going Stick is a more interesting alternative, since it also includes the bending motion or shear as well as the penetration.

Alternative methods to measure rotational peak shear with more sophisticated electronics make those measures more reliable, albeit more expensive. Lewis et al. [21] used such a device and found a weak linear relationship between rotational shear measured using the GWTT and longitudinal shear quantified using the OBST. The traction rotational tester described by ASTM F2333-04 did not show any relationship with the other tools mentioned above but detected differences in surfaces with higher loads. As a result, the simple devices used in this work are evaluated for their ability to make quantitative assessments accessible to more arena builders. It is important to understand the utility of these devices while recognizing the differences in loading between these devices and a horse.

### 1.4. Critical Factors

Based on prior work, the effect of moisture content, geotextile addition, and the addition of drainage all influence the functional properties of equestrian surfaces [14,32,33]. As modification that impact functional properties, and therefore performance, it is also reasonable to assume that they may have an impact on injuries and safety. Using silica sand-based test surfaces, lightweight standard measurement tools were used to characterize surfaces with different designs representing surfaces used in a wide range of facilities. While not replicating the full function of the more complex measurement of functional properties, the objective was to investigate the sensitivity of these tools to change in the design of the arena. These tools can be used for quality control when installing new arenas and may also be helpful when evaluating the consistency of maintenance by users. 

## 2. Materials and Methods

### 2.1. Study Design 

The design is a two-level experimental design with three factors (2^3^) (Table 1) with two repetitions (sixteen). Details of the construction are shown in Figure 1. Factors are the addition of geotextile: where no geotextile is added (G1) and with 2 $\frac{kg}{m^{2}}$ of geotextile chips added to the 10 cm deep surface (G2). The drainage package consists of a geotextile fabric laid over the limestone base, a geomesh layer and another geotextile fabric layer. The geotextile layers are used to avoid saturation of the geomesh with sand. The two conditions for the drainage layer are the absence of a drainage package (D1) and the incorporation of the drainage package (D2). Two gravimetric moisture contents (GMC %) were the lower at 11.16% ± 2.93 GMC (M1) or higher at 21.69% ± 3.90 GMC (M2). 

Sixteen boxes of 1 m × 1 m × 0.20 m were placed either on a compacted coarse base [34] or on a compacted coarse base with a draining package (Appendix A). Boxes were constructed following construction specifications of an equestrian surface manufacturer [35].

The dimensions of the test boxes were selected according to the Boussinesq equation to limit the edge effects [36]. Boxes with similar dimensions have been used by other investigators and are a part of established test protocols [33,37].

### 2.2. Test Boxes

Eight experiments with two repetitions (sixteen boxes) were set up over a compacted limestone base with a cross-slope of 0.7% (Figure 2). Sand was applied in two layers with each 5 cm layer compacted separately for a total of 10 cm of material over the base. Each layer was compacted using a 4 kg mass dropped three times from a height of 0.30 m onto an area of 0.20 m × 0.17 m. Compaction was performed in a similar way in all treatments. (Figure 3). 

The perimeter of the test boxes was defined by placing 200 cm long concrete tiles with a depth 20 cm and a width of 8 cm around the edge of the plot. The one square meter test boxes were divided by internal walls made with two rows of concrete tiles of 20 cm length, 10 cm depth and 8 cm width, which were further reinforced on the outside by identical concrete tiles offset by 10 cm to eliminate the gaps between the tiles. Sand was native to the area where the test boxes were located: the Equestrian Training Center of the Solaguayre Farm, in the town of Los Cardales, province of Buenos Aires, Argentina. Geotextile and mesh net were provided by a local equestrian surfaces builder and was consistent with local usage. The sand was tested using sieve and hydrometer tests to be 92.3% sand, 2.6% silt, and 5.1% clay (Testing by Racing Surfaces Testing Laboratory, Lexington, KY, USA). The particle size distribution of the sand was determined by ASTM D422 [38], silt and clay were tested by hydrometer [39]. The geotextile used in the boxes was 100% polyester based on FTIR testing (Lab Cor Materials, LLC, Seattle, WA, USA) [40]. Mineralogy of the sand was determined by X-ray diffraction analysis [41]. The bulk density was determined in accordance with ASTM D698 [42], for both sand and sand with the fiber reinforcement to determine the optimum moisture content for surface compaction (this is further described in the Appendix A).

Measurements of the VMC % was performed daily on a volumetric basis during compaction using TDR consistent with the ASTM D6870M-19 [43]. The gravimetric moisture content was also determined in the laboratory. Samples were taken from each box, it was weighed and placed in the oven at 65 °C for 48 h with a second weight taken for the dried material. The samples were re-weighed, and the moisture content was determined [44].

### 2.3. In Situ Measurement of Test Boxes

As described above, five in-situ measurements were performed using four measurement tools once the boxes were installed: the Going Stick Penetration (GSP) and Going Stick Shear (GSS) [45], the Impact Test Device (ITD) based on ASTM D5874, [25], the Rotational Peak Shear (RPS) device based on ASTM F2333 [26], and Volumetric Moisture Content (VMC), which is consistent with ASTM D6780 [43]. The operation and use of each of these devices was consistent with the applicable standards described below.

(a)The Going Stick Two Axis Sensor Probe

The Going Stick measures the penetration resistance and the resistance to rotation of the blade in the turf. The penetration shows the resistance to the penetration of the turf by the toe of the shoe and the rotation of the handle is a measure of the shear strength of the surface to a depth of 100 mm, which may be related to the rotation of the hoof into the surface and subsequent propulsion force from the horse (Figure 4).

The device has recently been proposed as an international standard measurement tool [45]. In addition to the two distinct measures of penetration and shear, the device also calculates an integrated measure of the two measured values, which is referred to as “the going index”. The going index is commonly used in Thoroughbred racing to describe the surface conditions [46]. The peak force required to press the probe into the surface and the peak torque applied to the handle to rotate the probe to 45 degrees are logged to memory in the device and are then used to calculate the going index. The torque is calculated from the strain gauges located 128 mm from the tip and the calculation of load assumes that the rotation of the device occurs around top of the plate of the probe, which penetrates into the surface [46,47].

The shear and penetration load values have also been converted from the measured values from the strain gauges to try to relate to the stress applied to the surface by using the area of the penetrating probe and the area of the side of the probe. Because the penetration probe is tapered and the device is not constrained horizontally, the load and torque applied to the top handle of the device has a complex relationship to the stress in the ground. However, making simplified assumptions, such as assuming a fixed point of rotation at the top of the surface where the probe penetrated into the footing, allows a load at a reference point to be calculated, which gives a reasonable quantifiable value for comparison.

The Going Stick was calibrated in a calibration fixture (Appendix B), with a mass loading the tip and the strain output was then converted to N and Nm for the penetration force and the applied torque respectively [47]. The use of a calibration with known loads and measurement locations reported in standard measurement units is consistent with the proposed standard test. The Going Stick software version 2.30 was used which does not average the values but saves the peak value of penetration and the shear. During the data collection and calibration, the Going Stick was set in flat mode.

(b)The Impact Test Device (ITD)

The surface hardness and resistance to compaction was measured using a custom impact test device (ITD) based on ASTM D5874-16 (Figure 5). The Clegg hammer is similar to the ITD, however it measures the maximum or the average deceleration of the mass over three to five drops [3]. The ITD, in contrast to the Clegg hammer, measures the deformation of the surface while still using a 2.25 kg mass dropped repeatedly inside a tube at each location from a height of 0.45 m (ITI: Impact Test Index). The principle is based on an impact velocity of a falling object with a known energy. The compaction of the surface and orientation of the tube results in variation in the distance traveled by the mass for the Clegg hammer, but the variation in distance is considered in the ITD. Unlike the various static and dynamic designs of penetrometers, the displacement on the ITD is measured for a larger impactor than the small square or conical impactors normally used in a penetrometer [31]. For the ITD, the distance to the impactor is measured with respect to a reference point at the top of the tube using a low-cost commercial laser distance measurement device (Model GLM 150 c, Bosch, Singapore). The cost of the system is a fraction of that of the commercial Clegg hammers and if produced locally would be within the means of nearly all arena builders and owners.

(c)Rotational Traction Tester (ASTM F2333-04)

The rotational shear strength was recorded using a torsional shear tester [26]. The device used is based on a modification of ASTM F2333-04 where the cleats on the disk are replaced by a horseshoe. The device, shown in Figure 6, is used in equine research to address the weaker correlation between rider perception and grip [21,38]. To pre-set the disk into the surface, a 30 kg weight was dropped 0.30 m along a shaft onto a second plate with a size 3 steel-studded horseshoe mounted on the bottom of the plate. The horseshoe included two 2.5 cm long cleats, which were tapered from 1.35 to 0.50 cm diameter on its abaxial face. The rotational peak shear (RPS) load was measured with a digital torque wrench with a range of 4–200 NM, and a precision of 0.08Nm (Model ARM602-4, ACDelco, Taiwan, China). To record the failure strength, the torque wrench is turned until it reaches the maximum load while keeping the plate flat on the ground. Five measurements were made for each box. In order to minimize operator variability, the same person performed all of the testing, which is consistent with best practice [48].

(d)Moisture Probe

Time domain reflectometry (TDR) is widely used to test volumetric moisture content (VMC %) as one of the primary factors required to achieve a consistent surface. The TDR moisture test probe (Spectrum Field Scout TDR-100 Aurora, IL, USA) was used with two measuring rods of 10 cm length. Five sample locations were measured at the inner part of each of the test boxes.

### 2.4. Statistical Analysis

Analysis of variance was performed using commercial statistical analysis software (Infostat version 2, Buenos Aires, Argentina). For comparison of marginal means, the Tukey test was performed. Values of *p* < 0.05 were considered statistically significant. 

Linear regression analysis between dependent variables and independent variables and Pearson’s coefficients of correlation were calculated to identify the degree of association between dependent variables. 

The proposed model for the testing of the boxes is:(1)$$Y_{\mathrm{ijk}}=µ+D_{i}+{\;M}_{j}+{\;G}_{k}+{\left(\mathrm{DM}\right)}_{\mathrm{ij}}+{\left(\mathrm{DG}\right)}_{\mathrm{ik}}+{\left(\mathrm{MG}\right)}_{\mathrm{jk}}+{\left(\mathrm{DMG}\right)}_{\mathrm{ijk}}+{\;e}_{\mathrm{ijkl}}$$
where D: Drainage _(1,2)_ M: Moisture _(1,2)_ G: Geotextile _(1,2)_

## 3. Results

Using the five measurements obtained from four devices, the effect of the different treatments of the boxes was shown to be significant for a number of different conditions: drainage (D), moisture (M) and the addition of geotextile (G). 

### 3.1. Going Stick

GSP was statistically significant for the drainage package (D) and for geotextile addition (G) (Table 2). GSP showed higher values with the D2 drainage system and with the addition of geotextile chips (G2) as well as being significant for the interaction between moisture content and the geotextile (MG) (Table 3, Figure 7). The treatments M1G2 and M2G2 were shown to have higher penetration values. GSS was statistically significant for the presence of the drainage package, D1:8.23 Nm ± 3.03 and D2: 5.73 Nm ± 2.18 (*p* < 0.05) and was positively associated with D (Table 4).

Linear regression analysis (Table 4) demonstrated that GSS was positively associated with D, and GSP was not significant.

### 3.2. Impact Test Device

The measurements made with the ITD were statistically significant for all three factors (Table 5) and for double interaction of drainage and geotextile (D × G) (Table 3, Figure 8). ITD was sensitive to both moistures treatments as shown in Table 2. The measured material displacement was higher when the drainage package was present; however, the geotextile can lead to similar displacement when the drainage package is absent. When the drainage package is present, the standard deviation is the lowest of both three other treatments (D1G1: 0.00983 ± 0.00323 vs D1G2:0.01376 ± 0.00304; D2G1:0.01272 ± 0.00215; D2G2:0.01472 ± 0.00278). ITD was positively associated with D and G (Table 4).

### 3.3. Rotational Peak Shear

The RPS was significant for geotextile addition (Table 6). The triple interaction of factors moisture, geotextile and presence of the drainage package is also significant (Table 3, Figure 9). RPS was also associated with G (Table 4). 

### 3.4. Moisture Probe

The Volumetric Moisture Content (VMC %) tested by TDR was significantly different for moisture treatments (Table 7). A regression to extract independent variables (G and D) for VMC produced an R^2^ of 0.80 and was significantly associated with M (*p* < 0.0021) (Table 4).

Correlations between devices showed that GSP was correlated with ITI (*p* < 0.0011; r = 0.38) and GSP was correlated with GSS (*p* < 0.0003; r = 0.42) with results shown in Table 8. 

## 4. Discussion

As previously noted, the aim of this study was to determine the ability to detect the moisture content, geotextile addition, and the presence of a drainage package on the measurements made on an equestrian arena using lower cost and readily available equipment portable tools. The design included silica sand locally sourced in Argentina and a drainage package commonly used as a base layer. The primary objective was to test the suitability of these tools to maintain consistency during and after construction of the surface. 

Although geotextile addition is a widespread practice, a comparison of identical sand without geotextile effects has not been investigated using these types of measurement tool. These data show that these tools can detect changes that result from the addition of geotextile materials. These attributes may be related to the firmness and cushioning of the surface. 

Although GSS is related to the slide of a longitudinal shear failure of the surface, results indicate that GSS has a weak linear relationship with the use of a drainage package (R^2^ = 0.23, 0.001) and could detect differences in longitudinal shear achieved in the experiment when drainage package is present (D1:8.23 Nm ± 3.03; D2: 5.73 Nm ± 2.18). The drainage package consists of two layers of geotextile with a geomesh in the middle; this arrangement has been used extensively for base materials in arenas. This type of geocell reinforcement has better performance than other types of geosynthetics due to its three-dimensional structure [49]. For base layer, the soil–reinforcement interface friction is lower than the soil–soil interface friction [50]. This may be the reason for lower longitudinal shear for the D2 treatment. 

Moisture content is generally an understood factor in the dynamic properties of equestrian surfaces [14,33]. When a horse gallops, the hoof exerts compressive and shear forces through the depth of the cushion. Ratzlaff [32] found that a moderate level of moisture was associated with lower levels of impact. In this experiment, bulk density behavior when geotextile was added was shown to be lower than without the geotextile. The same VMC (%) for both geotextile treatments (Figure A1 and Figure A2, Appendix A) led to different penetration resistance. The ITD was sensitive to both moisture treatments. The ITD is a measure of the displacement from the top to the bottom of the compaction of the cushion caused by dropping the mass. This displacement is increased at higher values of GMC. The M1 condition for GMC (11.16% ± 2.93) was outside of the range of the standard curve for the bulk density shown (Figure A2, Appendix A). The result is a very loose sand surface with fewer lubricated pores to facilitate reorientation of the sand particles. 

Higher ITI (0.01306 m ± 0.00314) at M2 of GMC represents the impact being lower at this moisture content. This is consistent with Ratzlaff [32] and may also indicate that energy rebound is present. ITI has a positive linear relationship with GSP (*p* = 0.38, *p* = 0.0011). 

The effect of geotextile on the response of the surface is related to relative motion between horseshoe and the surface. This resistance resulted in higher forces during a pivoting movement [29]. The GSP, ITD and RPS have proven to all be sensitive to this effect. The RPS has been addressed in previous work with weaker results [22]. In this paper a sensitivity is shown to the triple interaction of drainage package, moisture content and geotextile addition. Sliding of the hoof on the surface can either occur between the shoe and the surface, or within the material beneath the hoof depending on the characteristics of the surface and the design of the shoe [21]. Moisture content and the existence of the drainage package are thus able to affect the surface responses measured by RPS. Shear failure of the surface is detected by both the RPS and GSP measures when geotextile addition is tested, although R^2^ and r coefficient is not significant. The effective depth of the measurements is different and in fact the peak vertical force over the surface layers may be the controlling factor in these results. 

Moisture content, drainage package and geotextile treatments were shown to be significant for ITD. All three factors affected the distance so that the drop deformed the arena surface. Moisture and geotextile could achieve the uniformity that the use of the drainage package could not, as seen in the higher standard deviation. This result is consistent with the expected influence of the drainage package. 

Although increases in GSP could be observed in the combination of moisture and geotextile, the higher GSP showed that moisture has a similar result to the geotextile addition showed by ANOVA. Increases in GSP are higher when geotextile is added (M1G2 = 70.3%, M2G1 = 19.7% and M2G2 = 57.60%). 

The differences in the maximum bulk density of sand with and without geotextile may help with understanding this result. The bulk density varies with the amount of moisture with the maximum bulk density achieved at a specific moisture content. The moisture content where the soil particles are able to reorient while remaining in contact provides the greatest bulk density values. Curves for the bulk density of the sand and the sand with added geotextile showed differences in the moisture content range in which both systems remain stable to compaction (Figure A1 and Figure A2, Appendix A). Sand systems get higher bulk densities than sand with the addition of geotextile at both VMC% tested (Figure A2, Appendix A). The VMC% of the bulk density test cover the range of the present experiment (M1 = 10.20% ± 2.41 and M2: 25.08% ± 4.39 tested with TDR). This may suggest that geotextile addition with the compaction used in this experiment would respond as a low-density depth cushioning system. As Holt et al. [14] reported, low density surfaces have lower maximum peak loads and a lower peak acceleration on impact. In this experiment, hardness was represented by the lower distance achieved by the ITD when M1 was applied (high bulk density). Although resistance to penetration (GSP) is measured as a peak vertical load, the size of the probe and depth of penetration is very different to that of the horseshoe. 

The near surface properties measured with the RPS would not be expected to change with the addition of subsurface drainage. However, the drainage package atop limestone may be reducing the water flux rate and may also provide a more consistent interface. This lower rate of drainage may cause a more homogenous vertical distribution of water and avoid the loss of water shown by McInnes and Thomas in experimental turfgrass profile designs [51]. In the experimental boxes, a double geotextile layers with a mesh plastic was used. These results suggest that the mesh layer does not just serve as a separation layer but may also serve as a subsurface water storage. The mesh layer and the drainage layer may work together to provide more consistent water content both laterally in the boxes and vertically in the profile.

## 5. Conclusions

The effect and the interaction of moisture content, drainage and geotextile was tested using five measurements. Previous studies were confirmed with respect to the effect of moisture and geotextile addition that could both be detected with these simple instruments. The inclusion of a drainage package influenced surface measures such as shear and penetration measured using GS and the ITD. The effect of drainage on surface properties is likely to be related to the presence of both a vertical and horizontal water movement and the subsequent effect on consistency. The ability to test functional properties of the surface may be limited, however, due to the lower loads and the lower load rates. Simpler devices like GS, RPS or ITD may be best suited for quality control process in the construction of arenas rather than an evaluation of the suitability of the complete arena for performance and consistency. 

The ITD is closely related to methods widely use in sports field testing [52]. The ITD constructed for this experiment is cost effective and portable, which may justify further development. The ITD was the only device that could detect differences between the three factors included in this experiment. With more measurement locations and more accurate measurements of deflection, this device shows promise as a small, affordable tool for quality control. Other tools like the GS may be limited in applicability due to their design complexity and cost. The manufacturers of some subsurface mats used in certain equestrian disciplines, such as dressage, recommend profile depths less than 10 cm. The probe on the GS would be too long for use on those surfaces. While none of these devices directly correlate to a loading of the limbs of horses or the arena performance, many of the arenas have standard designs which, if properly installed, can provide reasonable functional parameters. As a result, when using these standard designs, a simple measurement device may suffice for quality control and can may play a useful role in the development of improved arena surfaces. Future work should consider alternative approaches that are also suited for monitoring the maintenance of the surface, to ensure that a high-quality arena will continue to provide a consistent performance over time. 

## Figures and Tables

**Figure 1 animals-11-02896-f001:**
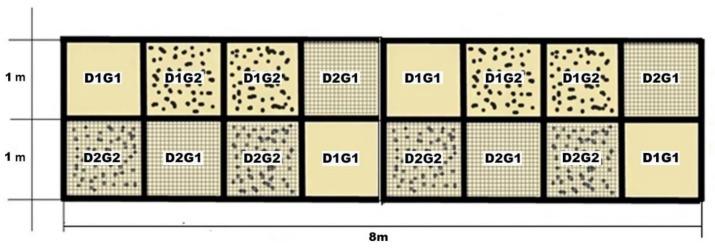
Organization of boxes distributed based on the top cushion material which consists of sand (G1) or sand with geotextile (G2) over limestone (D1) or over a drainage package (D2) along with combinations of both treatments (D1G1, D1G2, D2G1, D2G2).

**Figure 2 animals-11-02896-f002:**
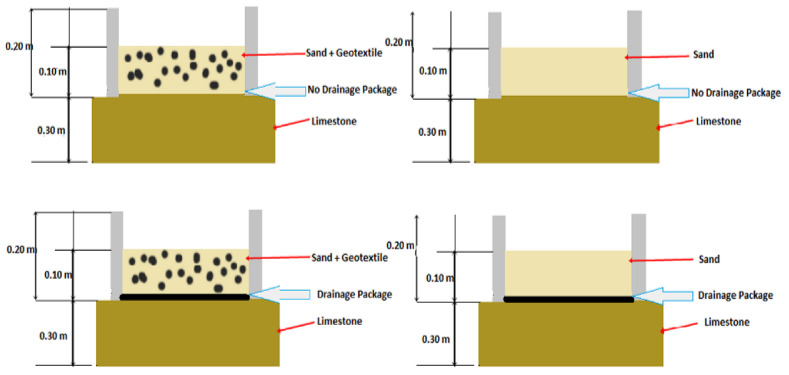
Scheme of box designs including the depth of the top cushion material consisting of sand (G1) or sand with geotextile (G2) over limestone (D1) or over drainage package (D2).

**Figure 3 animals-11-02896-f003:**
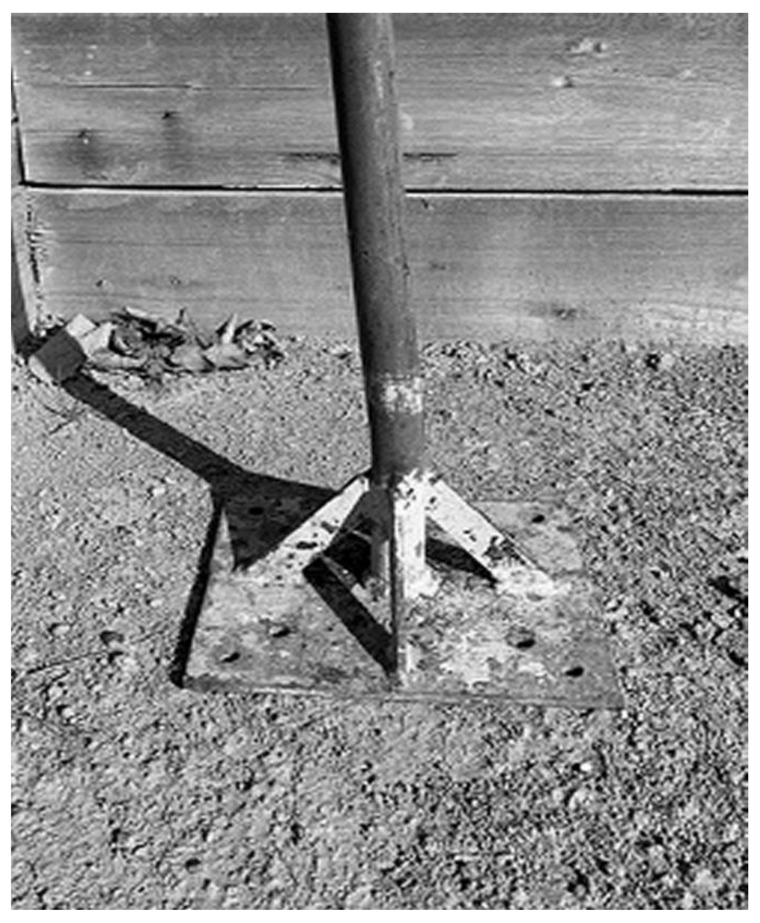
Image of the compaction tool with 4 kg mass dropped three times from a height of 0.30 m onto an area of 0.20 m × 0.17 m.

**Figure 4 animals-11-02896-f004:**
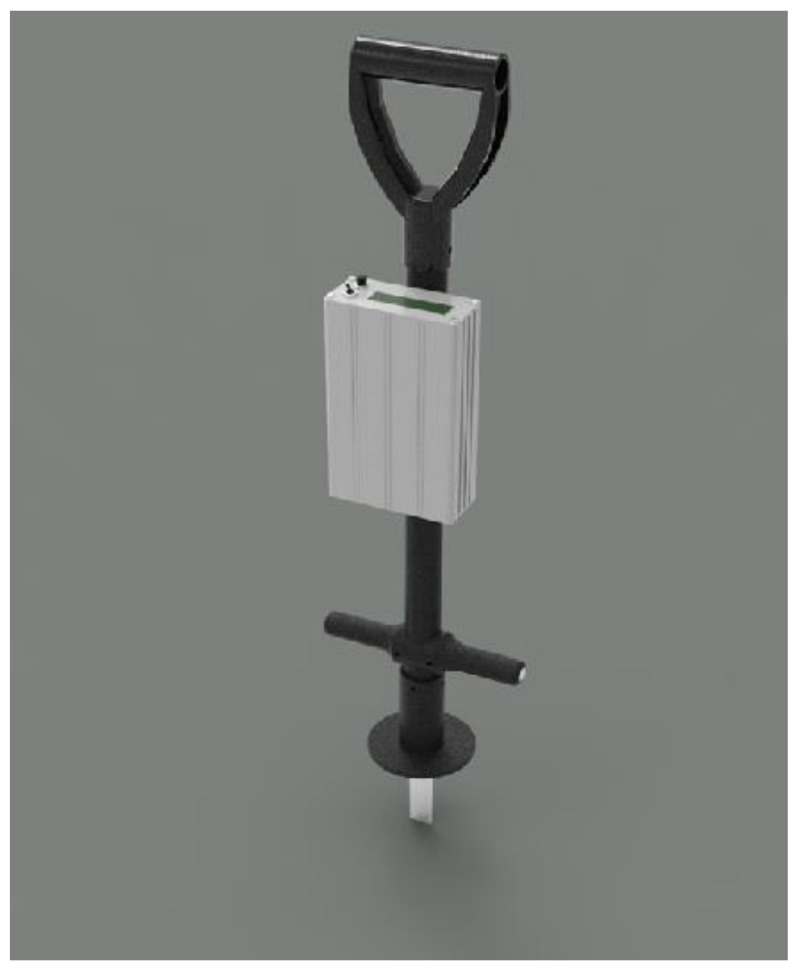
A drawing of the two axis commercial strain-gauge sensor, the Going Stick.

**Figure 5 animals-11-02896-f005:**
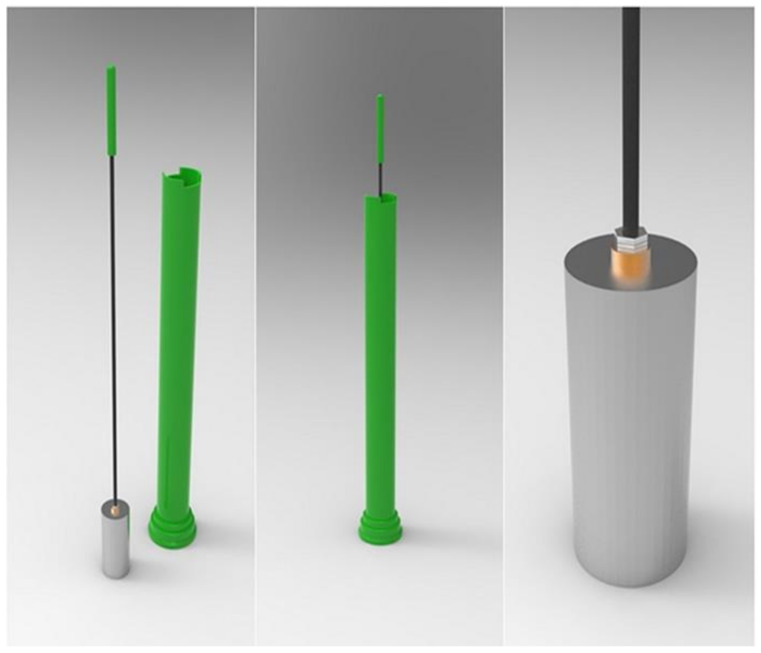
Custom impact test device (ITD) built to perform similarly to the device described in ASTM D5874-16.

**Figure 6 animals-11-02896-f006:**
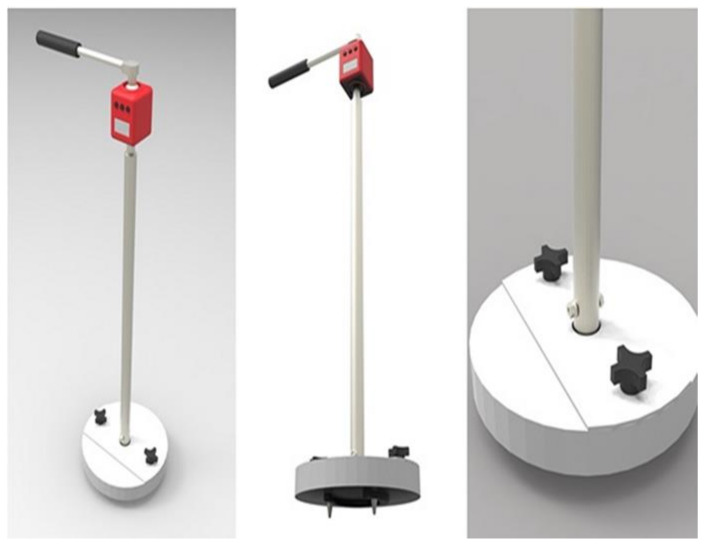
The rotational traction tester based on a device used for playing fields (ASTM F2333-04).

**Figure 7 animals-11-02896-f007:**
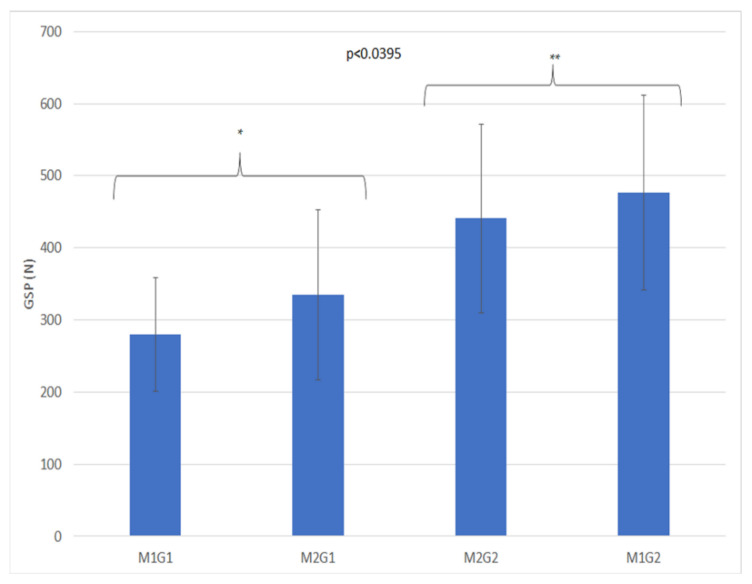
Mean of double interaction moisture x geotextile for GSP (in N) M 1: GMC was 11.16% ± 2.93 and treatment; M2: 21.60% ± 10.90; G1: without geotextile; G2: with 2 kg/m^2^ of geotextile. The stars (*, **) indicate significative statistically differences.

**Figure 8 animals-11-02896-f008:**
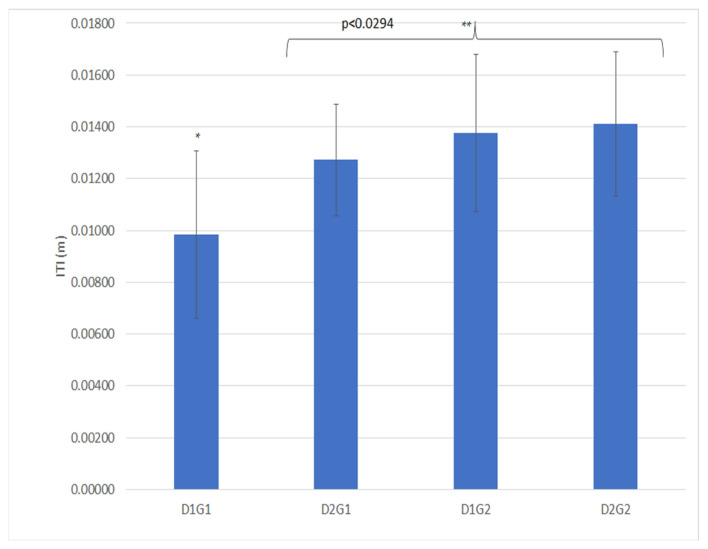
Double interaction of the drainage and geotextile for ITD (m). D1: no drainage; D2: drainage package; G1: without geotextile; G2: with 2 kg/m^2^ of geotextile. The stars (*, **) indicate significative statistically differences.

**Figure 9 animals-11-02896-f009:**
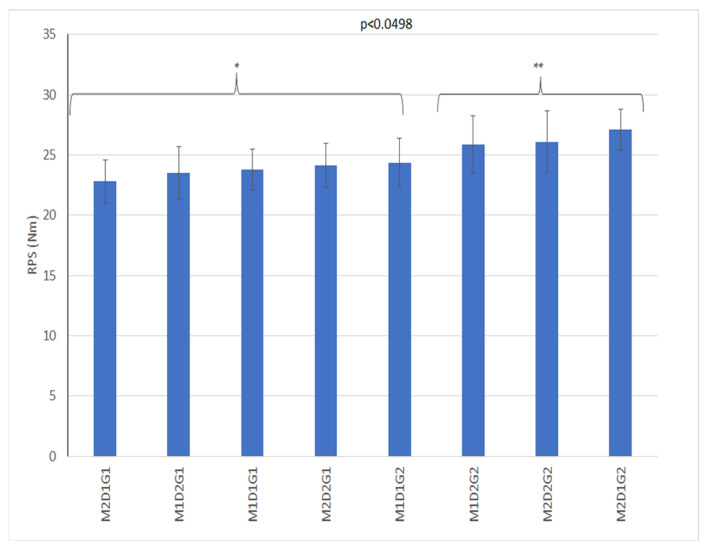
Means of triple interaction drainage, moisture, and geotextile for RPS (Nm). D1: no drainage; D2: drainage package; M1: GMC was 11.16% ± 2.93 and treatment; M2: 21.60% ± 10.90; G1: without geotextile; G2: with 2 kg/m^2^ of geotextile. The stars (*, **) indicate significative statistically differences.

**Table 1 animals-11-02896-t001:** Description of the 2^3^ experimental designs. G1: Without geotextile, G2: with 2 kg/m^2^ of geotextile; D1: without drainage package, D2: with drainage package; moisture: M1: Low gravimetric moisture content (11.16%± 2.93 GMC), M2: High gravimetric moisture content (21.69% ± 3.90 GMC) or high.

Factors		Drainage Package (D)			
		D1		D2	
		Moisture (M)			
		M1	M2	M1	M2
**Geotextile (G)**	G1	D1G1M1	D1G1M2	D2G1M1	D2G1M2
	G2	D1G2M1	D1G2M2	D2G2M1	D2G2M2

**Table 2 animals-11-02896-t002:** Statistics for the GSP and GSS (f and *p*-value), mean and standard deviation of GSP and GSS from Going Stick to three factors moisture, drainage and geotextile addition.

Variable	Moisture		Drainage		Geotextile	
	F	*p*	F	*p*	F	*p*
*GSP*	0.12	0.7307	9.68	0.0028	36.26	0.0001
Mean ± SD (N) (by levels)	M1	356.83 ± 165.35	D1	332.79 ± 162.53	G1	282.40 ± 120.87
	M2	368.47 ± 150.55	D2	394.16 ± 145.03	G2	444.56 ± 145.75
*GSS*	0.23	0.6343	10.39	0.0021	0.64	0.4271
Mean ± SD (Nm) (by levels)	M1	7.30 ± 2.57	D1	8.23 ± 3.033	G1	7.30 ± 2.91
	M2	6.63 ± 3.14	D2	5.73 ± 2.18	G2	6.65 ± 2.90

Tukey test Alpha = 0.05 *p* < 0.05.

**Table 3 animals-11-02896-t003:** F and *p*-values of interactions between factors: moisture × drainage, moisture × geotextile, drainage × geotextile and moisture × drainage × geotextile over the five variables (GSP, GSS, ITI, RPS, VMC).

Variable	Interactions							
	Moisture × Drainage		Moisture × Geotextile		Drainage × Geotextile		Moisture × Drainage × Geotextile	
	F	*p*	F	*p*	F	*p*	F	*p*
GSP	0.27	0.6059	4.42	0.0395	2.33	0.1319	3.09	0.0838
GSS	1.37	0.2465	0.26	0.6111	2.80	0.0996	1.22	0.2738
ITI	1.79	0.1860	0.01	0.9253	6.18	0.0156	2.27	0.1372
RPS	0.18	0.6724	0.16	0.6933	0.80	0.3754	4	0.0498
VMC	0.54	0.4668	0.01	0.9382	0.10	0.7507	0.49	0.4865

Tukey test Alpha = 0.05 *p* < 0.05.

**Table 4 animals-11-02896-t004:** Regression coefficients (R^2^), constant and signification of each linear model for every variable GSP, GSS, ITI, RPS and TDR.

Variable	R^2^	*p*			
		Const.	Moisture	Drainage	Geotextile
GSP	0.35	0.793	0.7234	0.0076 *	0.0001 *
GSS	0.23	<0.0001 *	0.5877	0.0008 *	0.2716
ITI	0.24	0.0194 *	0.1204	0.0203 *	0.0002 *
RPS	0.25	0.0001 *	0.5638	0.0558	0.0001 *
VMC	0.81	0.2224	<0.0001 *	0.6788	0.104

* Significant values at *p* < 0.05.

**Table 5 animals-11-02896-t005:** Statistic of ITI (f and *p*-value), mean and standard deviation of ITI to three factors Moisture, Drainage and Geotextile addition.

Variable	Moisture		Drainage		Geotextile	
	F	*p*	F	*p*	F	*p*
*ITI*	5.03	0.0284	4.75	0.0331	13.84	0.0004
Mean ±SD (m) (by levels)	M1	0.01208 ± 0.0032	D1	0.01208 ± 0.0037	G1	0.01148 ± 0.0030
	M2	0.01306 ± 0.0031	D2	0.01332 ± 0.0025	G2	0.01392 ± 0.0029

Tukey test Alpha = 0.05 *p* < 0.05.

**Table 6 animals-11-02896-t006:** Statistic of RPS (f and *p*-value), mean and standard deviation of RPS to three factors moisture, drainage and geotextile addition.

Variable	Moisture		Drainage		Geotextile	
	F	*p*	F	*p*	F	*p*
RPS	0.01	0.9356	3.59	0.0627	19.17	0.0001
Mean ±SD (Nm) (by levels)	M1	24.44 ± 2.26	D1	24.28 ± 2.28	G1	23.53 ± 1.91
	M2	24.74 ± 2.49	D2	24.94 ± 2.47	G2	25.68 ± 2.34

Tukey test Alpha = 0.05 *p* < 0.05.

**Table 7 animals-11-02896-t007:** Statistics for the VMC (f and *p*-value), mean and standard deviation of VMC for the three factors moisture, the presence of a drainage package and the addition of geotextile.

Variable	Moisture		Drainage		Geotextile	
	F	*p*	F	*p*	F	*p*
VMC	252.73	0.0001	0.28	0.5991	2.03	0.1594
Mean ±SD (%) (by levels)	M1	10.20 ± 2.41	D1	18.42 ± 8.5	G1	19.46 ± 8.06
	M2	25.08 ± 4.31	D2	18.99 ± 8.14	G2	17.95 ± 8.52

Tukey test Alpha = 0.05 *p* < 0.05.

**Table 8 animals-11-02896-t008:** Pearson correlation coefficients (r) of each variable GSP, GSS, ITI, RPS and TDR.

Variable	GSP		GSS		ITI		RPS		VMC	
	*p*	r	*p*	r	*p*	r	*p*	r	*p*	r
GSP	0.0001	1.00	0.0003 *	0.42	0.0011 *	0.38	0.1690	0.17	0.5444	0.07
GSS	0.0264	−0.27	0.0001 *	1	0.2583	−0.14	0.7407	−0.04	0.8792	0.05
ITI	0.0011	0.38	0.2583	−0.14	0.0001 *	1	0.2880	0.13	0.4140	0.10
RPS	0.1744	0.16	0.7407	−0.04	0.2880	0.13	0.0001 *	1	0.6385	0.06
VMC	0.55565	−0.07	0.8792	0.02	0.4140	0.10	0.6385	0.06	<0.0001 *	1

* Significant values at *p* < 0.05.

## Data Availability

Not applicable.

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
