# Peer review of "An Experimental Comparison of Simple Measurements Used for the Characterization of Sand Equestrian Surfaces"

_animals, 2021, doi:10.3390/ani11102896_

Round 1

Reviewer 1 Report

It was nice to see this data presented.  i have a couple of minor editorial comments

page 6 line 214 - consider changing done to performed

page 9 line 304  delete It was carried out... just start sentence at Linear....

page 10 please use past tense 

line 315  linear regression analysis (table 4) demonstrated that GSS was

line 315 replace is positively – with was positively

320 – replace is correlated to was correlated

Table 5 – pit is usually convention to use r2 for regression and r for correlation – this would be easier for the reader rather than 2 column both with the heading p

Page 15 line 489 delete manuscript type

Reviewer 2 Report

This is a good paper and important to the equine community in efforts to characterize surfaces in an efficient, affordable and consistent manner.

My strongest suggestion is to let the reader know exactly why this detailed paper/effort was conducted.  There is a lot covered in this paper, and one can get "lost" tying it all together. The last paragraph on page 2 (lines 66-77) should be written with a stronger emphasis on the need for mobile, affordable, instruments that can cover the essential characterization parameters that you discuss in the paper.  And, the Conclusion section should state conclusively which tool or tools are best for each specific test that you discuss, and if the ITD is the most useful overall, you should state so more strongly.

The grammar is overall fine, but commas missing in many paragraphs, I will state the lines below and other small corrections/suggestions:

Line

17-comma after sport

45-comma after result

51-comma after properties

55-should "limb' be "limbs"

59-replace "and use" with "using"

63-define "event"

72-comma after "ideally"

77-comma after "owners"

81-insert "surface" after "initial"

92-explain for the audience what the "current" tool are being used and why they are prohibitive versus the smaller tools you are recommending

96-comma after surface

103-consider other descriptor versus "analog"

110-comma after "tools"

112-comma after "standards"

123-break up sentence by putting period after track, and starting new sentence "They are also useful..."

127-delete "for this type of measurement"

131-delete "which"

142-write out "did not"

152-comma after "surfaces"

160-put 23 in parenthesis after "factors"

Figure 1: the G1=    , G2=   , etc pics are hard to see---consider using letters A,B,C,D, E instead and also embed these letters in the boxes.  Also enlarge the numerical dimensions--I can't see

175-should VMC be GMC?  if not, VMC should be defined in text (I don't think it was until later)

Figure 2 image needs to be larger--cannot read the numbers or text

243-missing period at end of sentence

249-explain what the "simplified assumptions" are

256-remove "of", after "values", suggest "but"

270-start of sentence consider "For" instead of "In"

307 and equation 1: explain why and where this equation originated from, if in the reference, explain briefly

4. Discussion: Suggest starting by stating "As cited earlier, "

376-expand sentence after "design" maybe stating "using smaller, affordable, readily available equipment" or something to that effect

Conclusion, last paragrah (476-489), see my comment at start of this review above.

Overall, this is a nice study!

Reviewer 3 Report

This is a very topical paper which tackles a significant issue within surface assessment; that of the effectiveness of more rudimentary equipment in assessing surfaces and surface change. 

To that end I recommend that this should be published.  Overall I feel the discussion, conclusions and methods used in the study are appropriate. 

Conversely, I recommend the authors look again at the way in which the work has been presented (particularly the results) and remove the inconsistencies which are apparent in the work.  I have attempted to highlight these below:

Whole document formatting error: Be consistent with your decimal places.  It varies a lot. 

Whole document formatting error: have used commas in some places and decimal points in others, please standardise.  

Line 4: Authors:  Superscript 3 on Dempsey

Line 26:   Quantitative measurements of performance parameters has have the potential to increase  

Line 44:  One space in front of  “ In addition the surface” (this could be a quirk of the pagination)

Line 125: Section 1.3 – strengthen the discussion of the disadvantages of these devices.  The reasons why devices like these are not used are because of a range of very good reasons.  You have captured the disadvantages in section 1.2 but not in 1.3.

Figure 1:  This is not clear - Label your squares with G1. G2 etc.  The purpose of this figure is demonstrate the layout of the different treatment combinations, it does not need to look like sand to the extent that you cannot tell the difference between some of the treatments. i.e. G1 and D1 are the same colour.

Line 173-176: this title needs to be clearer – remove extraneous wording to draw out the treatments – try something like this:

Table 1. Description of the 23 experimental design.

G1: Without geotextile, G2: with 2kg/m2 drainage

D1: without drainage package, D2: with drainage package described in methods.

M1: Low gravimetric moisture content (11,16%± 2.93 VMC), M2: High gravimetric moisture content (21,69 % ± 3,90 VMC) or high.

Line 319: Standardise your use of a decimal point (instead of a comma) when stating numbers i.e. p<0.0021 instead of p<0,0021 (This is repeated throughout the article)

Line 312 to 314: The key results in tables 2, 3 and 4 are not really described in the text.  There are 9 significant effects in table 2 and a similarly large number of interactions highlighted in table 3 – some attempt to describe /summarise them should be made.  I realise they are picked up later on when a review of each piece of equipment is undertaken but tables 2 and 3 are seemingly superfluous.

Line 324: Tukey not Tuckey.  Alpha not Alfa.

Line 329: In table 4 – remove the Linear model equations – they do not add to the understanding of the results.

The results section is confused.  The authors are trying to deliver some information about some relatively simple treatment effects but it is getting lost in the detail.

Table 2, 3 and 4 are seemingly not readily described in the initial stages of the analysis.  The focus of this study is the test equipment and how it detects changes in moisture, management and construction.  I would look at integrating the results in tables 2, 3 and 4 into sections 3.1, 3.2, 3.3 and 3.4 and therefore not relying on repeatedly referring back to these tables in each of the sections 3.1, 3.2, 3.3 etc.  This way, it allows the reader to focus their attention on responses and interactions of each piece of equipment (which is the point of the study).  It will require a re-think of the way the results are organised.   

Figures – add error bars to bar charts (either sd or se).  The experiment repeated and a graphical representation of the variation around the means would be useful.

Discussion

Lines 375-379:  Remove these lines.  This is a slightly unnecessary beginning as you have highlighted these elements already. Start the discussion at line 380.

Line 386: ‘These data’ not ‘This data’

Line 394: Remove ‘so called’

Line 399: Moisture content is a generally an understood factor in the dynamic properties of equestrian surfaces

Lines 399-411: This paragraph does not really make any conclusions.  It opens with some of the findings of Ratzlaff and compares them to an experimental finding – it requires more evaluative comments about the reasons why these findings are of interest.   

Line 443: Re-number your figures in your appendix to run concurrently with the figures in the main body.
